# State-driven Implicit Modeling

## Abstract

Implicit models are a general class of learning models that forgo the hierarchical layer structure typical in neural networks and instead define the internal states based on an "equilibrium" equation, offering competitive performance and reduced memory consumption. However, training such models usually relies on expensive implicit differentiation for backward propagation. In this work, we present a new approach to training implicit models, called State-driven Implicit Modeling (SIM), where we constrain the internal states and outputs to match that of a baseline model, circumventing costly backward computations. The training problem becomes convex by construction and can be solved in a parallel fashion, thanks to its decomposable structure. We demonstrate how the SIM approach can be further applied to parameter reduction and robust training by combining it with custom objective functions.

## 1 Introduction

Conventional neural networks are built upon a hierarchical architecture, where input information is processed through several recursive layers (Goodfellow et al., 2016). Canonical examples of this include standard feed-forward networks or convolutional networks (Krizhevsky et al., 2012; Simonyan & Zisserman, 2015). Recent work has proposed a more general perspective, where the internal states are implicitly defined through an "equilibrium" equation (Bai et al., 2019; Chen et al., 2018; El Ghaoui et al., 2021), allowing for loops in the model's computational graph. As illustrated in Bai et al. (2020); Gu et al. (2020), implicitly-defined models are able to match state-of-the-art performance of explicitly-defined models on several tasks. In fact, the implicit framework is a more general model with greater capacity to possibly model novel architectures and prediction rules for deep learning that are not necessarily tied to any notion of "layers."

The forward pass of an implicit model usually relies on solving an algebraic equation using methods such as fixed-point equations (El Ghaoui et al., 2021), ODE solvers (Dupont et al., 2019), or root-finding methods (Bai et al., 2020). The backward pass involves differentiating through the implicit equation, which usually relies on expensive black-box solvers, projected gradient descent, or approximate gradients (Geng et al., 2021). The costly backward computation remains a challenge in the training and evaluation of implicit models. In this work, we develop a novel method to circumvent computing the backward pass. We start from a baseline model (*e.g.*, a pre-trained layered neural network) and constrain the states and outputs of the implicit model to match those baseline states. The SIM training problem is strictly feasible and convex by construction and thus can be solved efficiently, bypassing the expensive implicit differentiation. Additionally, the method is very scalable: it can be implemented in parallel provided that the objective is decomposable across its internal state, which is usually the case.

We find that with our approach, the number of training samples required to efficiently and effectively train an implicit model is significantly reduced. For example, using 20%-30% of total training data is enough to train an implicit model on CIFAR-100 dataset. Our method can also be combined with additional objectives such as parameter reduction or improving robustness, making it a versatile training scheme.

Our main contributions are summarized as follows:

- We propose a general **S**tate-driven **I**mplicit **M**odeling (SIM) training scheme to efficiently learn an implicit model by matching the internal state and outputs of a baseline model without expensive implicit differentiation.
- We present simple ways to obtain the internal state and outputs from either a standard (layered) neural network or an implicit model.
- We introduce an efficient parallel convex training algorithm for the SIM training problem.
- We demonstrate the efficacy of the SIM algorithm on parameter reduction and robustness.

Our experimental results display a competitive performance of our method on both parameter reduction and improving robust training, motivating future directions of research in this new class of learning model.

## 2 Preliminaries

**Notations.** Throughout the paper, we use $n, m, p, q$ to denote the number of internal states, the number of input samples, the dimension of input vectors, and the dimension of output vectors, respectively. For a matrix $V$, $|V|$ denotes its absolute value (*i.e.* $|V|_{ij} = |V_{ij}|$); $\|V\|_0$ is its cardinality, *i.e.*, the number of non-zero entries of $V$; $\|V\|_\infty$ is the max-row-sum matrix operator norm; $\|V\|_F$ is the Frobenious norm. Finally, $\lambda_{\mathrm{pf}}(M)$ denotes the *Perron-Frobenius (PF) eigenvalue* of a square non-negative matrix $M$ (Berman & Plemmons, 1994).

**Assumption 2.1** (component-wise non-expansive)**.** A function $\phi$ is **co**mponent-wise **n**on-**e**xpansive (CONE) if
$$\forall\, u, v \in \mathbb{R}^n \;:\; |\phi(u) - \phi(v)| \le |u - v|,$$
with inequality and absolute value taken component-wise.

We are given a data set with input matrix $U \in \mathbb{R}^{p \times m}$ and output matrix $Y \in \mathbb{R}^{q \times m}$, where each column represents an input or output vector. An implicit model consists of an equilibrium equation in a "state matrix" $X \in \mathbb{R}^{n \times m}$ and a prediction equation:

$$X = \phi(AX + BU) \quad \text{(equilibrium equation)} \tag{1a}$$

$$\hat{Y}(U) = CX + DU \quad \text{(prediction equation)} \tag{1b}$$

where $\phi : \mathbb{R}^{n \times m} \to \mathbb{R}^{n \times m}$ is a nonlinear activation that is strictly increasing and satisfies Assumption (2.1), such as ReLU, tanh, or sigmoid. While the above model seems very specific, it covers as special cases most known architectures arising in deep learning. Matrices $A \in \mathbb{R}^{n \times n}$, $B \in \mathbb{R}^{n \times p}$, $C \in \mathbb{R}^{q \times n}$ and $D \in \mathbb{R}^{q \times p}$ are model parameters. In equation (1a), the input feature matrix $U \in \mathbb{R}^{p \times m}$ is passed through a linear transformation by weight matrix $B$ and the internal state matrix $X \in \mathbb{R}^{n \times m}$ is obtained as the fixed-point solution to equation (1a). The output prediction $\hat{Y}$ is then obtained by feeding the state $X$ through the prediction equation (1b). The structure is illustrated in Figure 1, where the "pre-activation" and "post-activation" state matrices $Z, X$ are shown; in those matrices, each column corresponds to a single data point.

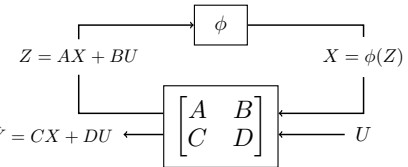

Figure 1: A block-diagram view of an implicit model, where $Z$ is the pre-activation state "before" passing through the activation function $\phi$ and $X$ is the post-activation state "after" passing through $\phi$.

The forward pass of an implicit model relies on the fixed-point solution of the underlying equilibrium equation, while a backward pass requires one to differentiate this equation with

respect to the model parameters $(A, B, C, D)$. The solution to the equilibrium equation (1a) does not necessarily exists nor be unique. We say that an equilibrium equation with activation map $\phi$ is well-posed if the following *well-posedness* condition is satisfied (El Ghaoui et al., 2021).

**Definition 2.2** (well-posedness). The $n \times n$ matrix $A$ is said to be well-posed for $\phi$ if, for any $b \in \mathbb{R}^n$, the solution $x \in \mathbb{R}^n$ of the following equation $x = \phi(Ax + b)$ exists and is unique.

**Scaling the network.** Consider a standard layer-based neural network $\mathcal{N} : \mathbb{R} \times \mathbb{R} \to \mathbb{R} \times \mathbb{R}$ with activation $\phi$ that satisfies Assumption (2.1) and maps input feature matrix $U \in \mathbb{R}^{p \times m}$ to outputs $Y = \mathcal{N}(U)$ via hidden layers. As shown in El Ghaoui et al. (2021), for such networks, there exists an equivalent implicit model, $(A_{\mathcal{N}}, B_{\mathcal{N}}, C_{\mathcal{N}}, D_{\mathcal{N}}, \phi)$ as in (1). Without loss of generality, we may re-scale the original weight matrices of $\mathcal{N}$ to obtain a strongly well-posed implicit model, $(A'_{\mathcal{N}}, B'_{\mathcal{N}}, C'_{\mathcal{N}}, D'_{\mathcal{N}}, \phi)$, by Theorem (2.4), in the sense that $\|A'_{\mathcal{N}}\|_\infty < 1$. This result also allows us to consider the convex constraint $\|A\|_\infty < 1$ as a sufficient condition as opposed to the non-convex PF sufficient condition, in light of the bound $\lambda_{\mathrm{pf}}(A) \leq \|A\|_\infty$.

**Theorem 2.3** (PF sufficient condition for well-posedness). *Assume that $\phi$ satisfies Assumption (2.1). Then, $A$ is well-posed for any such $\phi$ if $\lambda_{pf}(|A|) < 1$. Moreover, the solution $x$ of equation (1a) can be computed via the fixed point iterations $x \to \phi(Ax + b)$, with initial condition $x = 0$.*

**Theorem 2.4** (Rescaled implicit model). *Assume that $\phi$ is CONE and positively homogeneous, i.e., $\phi(\alpha x) = \alpha \phi(x)$ for any $\alpha \geq 0$ and $x$. For a neural network $\mathcal{N}$ with its equivalent implicit form $(A_{\mathcal{N}}, B_{\mathcal{N}}, C_{\mathcal{N}}, D_{\mathcal{N}}, \phi)$, where $A_{\mathcal{N}}$ satisfies PF sufficient condition for well-posedness of Theorem (2.3), there exists a linearly-rescaled equivalent implicit model $(A'_{\mathcal{N}}, B'_{\mathcal{N}}, C'_{\mathcal{N}}, D'_{\mathcal{N}}, \phi)$ with $\|A'_{\mathcal{N}}\|_\infty < 1$ that gives the same output $\hat{Y}$ as the original $\mathcal{N}$ for any input $U$.*

The proofs of Theorem (2.3) and (2.4) are given in the appendices.

## 3 STATE-DRIVEN IMPLICIT MODELING

The **S**tate-driven **I**mplicit **M**odeling (SIM) framework trains an implicit model with a constraint: it should match both the state $X$ and outputs $\hat{Y}$ of another "baseline" (implicit or layered) model, when the same inputs $U$ are applied. For a given baseline model, the state matrix $X$ can be obtained by running a set of fixed-point iterations (if the baseline is implicit), or a simple forward pass (if the baseline is a standard layered network). In both cases, we can extract the pre-activation state matrix $Z$, such that the post-activation state matrix satisfies $X = \phi(Z)$. Each column of matrices $Z$ and $X$ corresponds to a single data point; when the baseline is a layered network, these matrices are constructed by stacking all the intermediate layers into a long column vector, where the first intermediate layer is at the bottom and the last intermediate layer is on top.

We give a simple example of how to construct $X, Z$ from a 3-layer fully-connected network of the form:
$$\hat{y}(u) = W_2 x_2, \ x_2 = \phi(W_1 x_1) \ x_1 = \phi(W_0 x_0), \ x_0 = u,$$
where $u$ is a single vector input. For notational simplicity, we exclude the bias terms, which can be easily accounted for by considering the vector $(u, 1)$ instead of $u$. Each column of $Z$ and $X$ corresponds to the state from a single input. The column $z$ is formed by stacking all the intermediate layers before passing through $\phi$ and the column $x$ is formed by stacking all the intermediate layers after passing through $\phi$:
$$z = \begin{pmatrix} W_1 x_1 \\ W_0 x_0 \end{pmatrix}, \quad x = \phi(z) = \begin{pmatrix} x_2 \\ x_1 \end{pmatrix}.$$

In this example, we can easily verify that its equivalent implicit from is as follows:
$$\left( \begin{array}{c|c} A & B \\ \hline C & D \end{array} \right) = \left( \begin{array}{cc|c} 0 & W_1 & 0 \\ 0 & 0 & W_0 \\ \hline W_2 & 0 & 0 \end{array} \right).$$

For a more complicated network, finding an equivalent implicit form may be a non-trivial task. The SIM framework allows us to consider any baseline deep neural networks without ever having to address this challenge: we simply need to extract the pre- and post-activation state matrices.

With matrices $X, Z$ fixed, we now consider the training problem, where the model parameters are encapsulated in a partitioned matrix $M \in \mathbb{R}^{(n+q) \times (n+p)}$ and we define $\tilde{Y}, \tilde{U}$ as follows:

$$M := \begin{pmatrix} A & B \\ C & D \end{pmatrix}, \ \tilde{Y} := \begin{pmatrix} Z \\ \hat{Y} \end{pmatrix}, \ \tilde{U} := \begin{pmatrix} X \\ U \end{pmatrix}$$

The condition $\tilde{Y} = M\tilde{U}$ *characterizes* the implicit models that match both the state and outputs of the baseline model. We then solve a *convex problem* to find another well-posed model, with a desired task in mind, with the matching condition $\tilde{Y} = M\tilde{U}$:

$$\min_{A,B,C,D} \quad f(A, B, C, D) \tag{2a}$$

$$\text{s.t.} \quad Z = AX + BU, \tag{2b}$$

$$\hat{Y} = CX + DU, \tag{2c}$$

$$\|A\|_\infty \leq \kappa. \tag{2d}$$

Here, $f$ is an user-designed objective function chosen for a desired task, such as encouraging sparsity, and $\kappa \in (0, 1)$ is a hyper-parameter. Note that for a given input matrix $U \in \mathbb{R}^{p \times m}$, we have generically $U^T U \succ 0$, when $m > p$. The matrix equation $\tilde{Y} = M\tilde{U}$ involves $(n+p)m$ scalar equations in $(n + p)(n + q)$ variables, it is thus natural to require that $n > m - p$, which is generally true for over-parameterized models.

The *state-matching constraint* (2b) ensures that the implicit model determined by the weight matrices $A, B, C, D$ achieves the same representational power as the baseline model $\mathcal{N}$ by having the same internal state. The *outputs-matching constraint* (2c) ensures that the model achieves the same predictive performance by obtaining the same predictions as $\mathcal{N}$. Finally, the *well-posedness constraint* (2d) is added to ensure that the well-posedness condition is satisfied.

For a given baseline layered neural network model, we can always rescale the state matrices $X, Z$ by Theorem (2.4), so that the problem is strictly feasible. Denoting by $W_\ell$ the network's matrix corresponding to layer $\ell$, we divide it by the largest max-row-sum norm of the weights among all the layers:

$$W_\ell' = \frac{W_\ell}{\gamma \cdot \max_\ell \|W_\ell\|_\infty}, \ \ell \in [L], \ \gamma > 1,$$

where $L$ is the total number of layers and $\gamma$ is a scaling factor. The corresponding state matrices $X, Z$ will then be appropriately rescaled after running a single forward pass.

### 3.1 STATE-DRIVEN TRAINING PROBLEM

SIM is a general training scheme and various kinds of tasks can be achieved by including an appropriately designed objective and setup. We show two such possibilities: one aims for improved sparsity and the other for improved robustness.

**Training for sparsity.** To learn a sparse implicit model, we consider the SIM training problem where we sparsify the weight matrix $M$ by minimizing its cardinality, while satisfying $\tilde{Y} = M\tilde{U}$:

$$\min_M \quad \|M\|_0 \ : \ (2b)\text{-}(2d). \tag{3}$$

In general, solving the optimization problem (3) directly is not computationally efficient, and therefore a common alternative is to consider a convex relaxation. We consider the *perspective relaxation* that is a significantly stronger approximation (Frangioni & Gentile, 2006; Atamtürk & Gómez, 2019; Atamturk et al., 2021) than the popular $\ell_1$-norm relaxation,

and has recently been used for pruning neural networks (Cacciola et al., 2022). This leads to the following training problem:

$$\min_{M,t} \quad \sum_{ij} \frac{M_{ij}^2}{t_{ij}} + \sum_{ij} t_{ij} \ : \ \text{(2b)-(2d)}, \ t_{ij} \in [0,1]. \tag{4}$$

The perspective terms $\frac{M_{ij}^2}{t_{ij}}$ are typically replaced with auxiliary variables $s_{ij}$ along with rotated cone constraints $M_{ij}^2 \le s_{ij} \cdot t_{ij}$ (Aktürk et al., 2009), leading to a second-order cone problem:

$$\min_{M,t,s} \quad \alpha \sum_{ij} s_{ij} \ : \ \text{(2b)-(2d)}, \ t_{ij} \in [0,1], M_{ij}^2 \le s_{ij} \cdot t_{ij}, \ s_{ij} \ge 0, \tag{5}$$

where $\alpha$ is a hyper-parameter that controls the degree of sparsity. Problem (5) can be easily solved with conic quadratic solvers.

**Training for robustness.** To promote robustness, we consider regularizing the $\ell_1$-norm of the weight matrix $M$. The use of norm-based regularization (*e.g.* $\ell_2$ or $\ell_1$-norm) for training neural networks has been widely adopted. It has also been shown that there exists an intrinsic relationship between regularizing the $\ell_1$-norm of the weight matrices and their robustness against $\ell_\infty$-bounded perturbations (Guo et al., 2018; Alizadeh et al., 2020). The set of $\ell_\infty$-bounded perturbations yields the worst-case scenario since it includes all other $\ell_p$-bounded perturbations. Controlling the $\ell_1$-norm, therefore, guarantees robustness to $\ell_\infty$-perturbations and thereby to all other $\ell_p$-bounded perturbations. Note that we are minimizing the vectorized $\ell_1$-norm of $M$, i.e. $\sum_{ij} |M_{ij}|$, instead of the matrix operator norm. The resulting training problem:

$$\min_{M} \quad \sum_{ij} |M_{ij}| \ : \ \text{(2b)-(2d)}, \tag{6}$$

is *convex* and can be solved efficiently by a standard optimization solver.

**Relax state and outputs matching.** For the state-matching and output-matching constraints (2b) and (2c), it is not necessary to insist on an exact match. This allows us to relax (2b) and (2c) by introducing penalty terms into the objective function:

$$\min_{M} \quad f(M) + \lambda_1 \|Z - (AX + BU)\|_F^2 + \lambda_2 \left\|\hat{Y} - (CX + DU)\right\|_F^2 \ : \ \text{(2d)}, \ \mathcal{C}, \tag{7}$$

where $f$ and $\mathcal{C}$ are a user-defined objective function and constraint set on model parameters. $\lambda_1$ and $\lambda_2$ are hyper-parameters that control the degree of state- and output-matching.

**Parallel training.** The SIM training problem can be decomposed into a series of parallel, smaller problems, each involving a single row, or a block of rows, if $f$ is decomposable. This is usually the case, including in the sparsity and robustness examples seen before. For a single row $(a^T, b^T)$ of $(A, B)$, and with $z^T$ the corresponding row in $Z$, the problem takes the form of a basis pursuit problem:

$$\text{Find vectors } a,b \ : \ z = \begin{pmatrix} X^T & U^T \end{pmatrix} \begin{pmatrix} a \\ b \end{pmatrix}, \|a\|_1 \le \kappa. \tag{8}$$

where $\|a\|_1 \le \kappa$ is the well-posedness condition since $\|A\|_\infty$ is separable in terms of rows. The problem of finding $C, D$ is independent of that relative to $A, B$ and takes the same form as problem (8) without the well-posedness condition:

$$\text{Find vectors } c,d \ : \ \hat{y} = \begin{pmatrix} X^T & U^T \end{pmatrix} \begin{pmatrix} c \\ d \end{pmatrix}. \tag{9}$$

The decomposibility is applicable to the perspective relaxation and the $\ell_1$-norm objective that we consider, with appropriate constraint set $\mathcal{C}$. The parallel SIM training algorithm is summarized in Algorithm 1. More implementation details on parallel training can be found in the appendices.

---

**Algorithm 1** Parallel **S**tate-driven **I**mplicit **M**odeling (SIM)

---

**Input**: Input feature matrix $U$; A standard (layered) neural network or a pre-trained implicit model $\mathcal{N} : \mathbb{R} \times \mathbb{R} \to \mathbb{R} \times \mathbb{R}$; Well-posedness hyper-parameter $\kappa$.
**Design choices**: Convex minimization objective $f$; Convex constraint set $\mathcal{C}$; Hyper-parameters for $f$.
**Output**: Weight matrices $A$, $B$, $C$, $D$.

1: **if** $\mathcal{N}$ is a standard (layered) neural network **then**
2:     Run a single forward pass on $\mathcal{N}$ with $U$ to obtain outputs $\hat{Y}$, *i.e.*, $\hat{Y} = \mathcal{N}(U)$.
3:     Collect all intermediate layers before and after passing through the activation $\phi$.
4:     Construct $Z$ and $X$ by stacking all intermediate layers.
5: **else**
6:     Run fix-point iteration until converge for (1a) to obtain $X$ and $Z$.
7:     Run prediction equation (1b) to obatin $\hat{Y}$.
8: **end if**
9: Put $X, U$ into shared memory.

**begin parallel training**
10: Let $A \leftarrow \mathbf{0} \in \mathbb{R}^{n \times n}, B \leftarrow \mathbf{0} \in \mathbb{R}^{n \times p}$
    $C \leftarrow \mathbf{0} \in \mathbb{R}^{q \times n}, D \leftarrow \mathbf{0} \in \mathbb{R}^{q \times p}$.
11: Distribute rows of $Z$ or $\hat{Y}$ to each processor.
12: **for** each processor, in parallel **do**
13:     Solve one of the following convex optimization problems (or the relaxed version shown in Eq. (7)):

$$\min_{a,b} f(a,b) \ : \ z = \begin{pmatrix} X^T & U^T \end{pmatrix} \begin{pmatrix} a \\ b \end{pmatrix}, \ \|a\|_1 \leq \kappa, \ \mathcal{C}$$

$$\min_{c,d} f(c,d) \ : \ \hat{y} = \begin{pmatrix} X^T & U^T \end{pmatrix} \begin{pmatrix} c \\ d \end{pmatrix}, \ \mathcal{C}$$

14: **end for**
15: Update rows of $A, B$ or $C, D$

---

## 4    NUMERICAL EXPERIMENTS

We demonstrate the effectiveness of SIM in training an implicit model from a standard (layered) network for the following two tasks: 1) parameter reduction and 2) robust training. We test our method on both image and text classification datasets. These experiments were performed on 60 INTEL XEON processors and solved using MOSEK (ApS, 2022) optimization solvers. The test set performance is reported. More details on the numerical experiments can be found in the appendices.

### 4.1    TRAINING FOR PARAMETER REDUCTION

In these experiments, we solve the SIM training problem using the perspective relaxation and $\ell_1$-norm objectives with relaxed state and output matching penalties as in problem (7), allowing us to control the trade-off between parameter reduction, state-matching, and outputs-matching through hyper-parameters. Throughout the rest of the paper, we use the following hyper-parameters for experiments if not explicitly specified: $\kappa = 0.99$ for well-posedness condition, $\lambda_1 = \lambda_2 = 0.1$ for state- and output-matching condition, $\alpha = 1 \times 10^{-3}$ for sparsity. To evaluate the performance, we define sparsity as follows:

$$\text{Sparsity (\%)} := \left( 1 - \frac{\|M\|_0}{\|\mathcal{N}\|_0} \right) \times 100$$

where $\| \cdot \|_0$ is the total number of non-zero parameters.

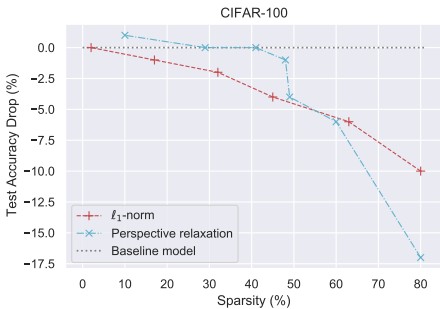 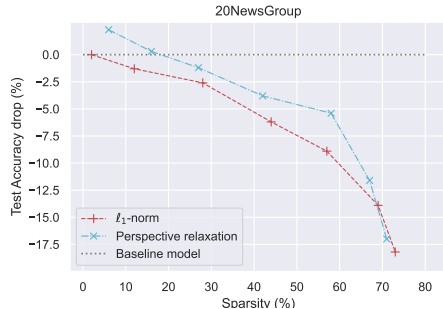

Figure 2: Trade-off curve for sparsity and test accuracy drop for CIFAR-100 and 20NewsGroup datasets. Perspective relaxation performs better than $\ell_1$-norm at parameter reduction.

Our experiments were evaluated on both image classification and text classification tasks. We use CIFAR-100 (Krizhevsky, 2009), a 32×32 colour images in 10 classes, and 20NewsGroup dataset[1], a public dataset consisting of 20 different newsgroups. For CIFAR-100, we use a ResNet-20 convolutional neural network, denoted as $\mathcal{N}_{res}$, to construct the state matrices $X, Z$ and the outputs $\hat{Y}$. We follow the hyper-parameter settings in Devries & Taylor (2017) with a mini-batch size of 128. The model is trained for 200 epochs, with a 75.8% test performance. For 20NewsGroup dataset, we use a DistilBERT model[2] (Sanh et al., 2019), denoted as $\mathcal{N}_{bert}$, to construct the state matrices and outputs. The model is for trained 100 epochs, with 82% F1 score on the test set[3].

Figure 2 shows the trade-off curve for sparsity and test accuracy drop for both datasets. The experiments show that using perspective relaxation as objective yields a 41% and 17% reduction of the parameters with no accuracy drop for CIFAR-100 and 20NewsGroup datasets respectively. Although $\ell_1$-norm is a more widely used objective for learning sparse models, it is less effective as compared to the perspective relaxation, which is a stronger relaxation for $\ell_0$. Moreover, the perspective relaxation further increases the test performance on both dataset while reducing around 10% and 6% of the parameters.

In solving the problem (7), the input matrix $U \in \mathbb{R}^{p \times m}$ does not have to be the full training dataset. We test how many numbers of samples are required to sufficiently train a sparse implicit model. Figure 3 shows the effect of the number of samples on sparsity for both datasets. A higher percentage of total training samples means higher $m$ for input matrix $U$. Negative sparsity means that the trained implicit models contain more parameters than the baseline model. For CIFAR-100, when learned with perspective relaxation, we can reduce 10% of the parameters by using only 20% of the total training data without sacrificing the test accuracy. We observe a similar result for 20NewsGroup data, where our method achieves 17% fewer parameters using less than 15% of the total training data with no test accuracy loss. The results indicate that the state matrix $X$ is a high-quality representation that captures a large number of the underlying information, and hence it is sufficient to train a model with significantly fewer training samples. Although the state matrix $X$ is obtained from a standard neural network, we see that in Figure 2 we are still able to increase the test performance further with fewer parameters using implicit models. This suggests that implicit models could provide a better representation as compared to a standard layered neural network.

Finally, we compare our method (denoted as SIM) on CIFAR-100 with other parameter reduction methods, including SSS (Huang & Wang, 2018), SPR (Cacciola et al., 2022), and MLA (Hu et al., 2019). SSS and SPR both formulate the task as a sparse regularized optimization problem similar to ours, where SSS uses $\ell_1$-relaxation and SPR uses perspective relaxation. MLA considers aligning the semantic information of the intermediate outputs and overall performance of the baseline model and the sparse model by introducing a feature and

---

[1] http://qwone.com/~jason/20Newsgroups/
[2] https://huggingface.co/docs/transformers/model_doc/distilbert
[3] Both models are trained on a single NVIDIA TITAN V GPU.

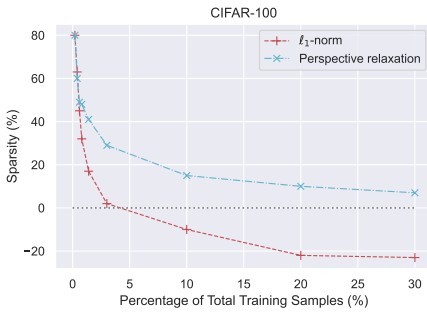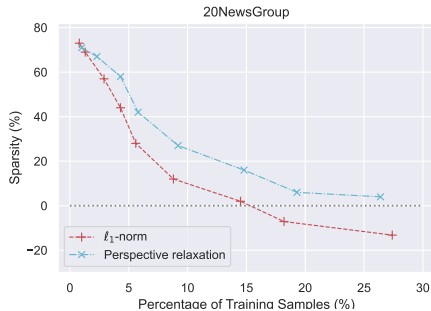

Figure 3: The effect of the number of samples on sparsity for FashionMNIST and CIFAR-100 datasets. Perspective relaxation outperforms the $\ell_1$-norm at parameter reduction for the same number of samples.

semantic correlation loss and a classification loss, similar to our state- and outputs-matching conditions. SSS and SPR both use ResNet-20 and MLA uses ResNet-18. We report the results of each method as they were reported in the original papers. For SSS, we use the results reported by Cacciola et al. (2022), where the data points are approximated from figures 3(c) of the paper and denoted as P1 and P2. Table 1 shows that SIM achieves less accuracy drop while reducing a larger amount of parameters. With around 30% of sparsity, SIM has a much lower accuracy drop as compared to MLA and maintains a similar accuracy drop as SPR. With around 45% of sparsity, SIM outperforms all three methods with a lower accuracy drop and more reduced parameters. It is also likely that additional parameter tuning may lead to more competitive results.

Table 1: Comparison with other parameter reduction methods on CIFAR-100.

| Method | Setting | Acc. Drop (%) | Sparsity (%) |
|--------|---------|---------------|--------------|
| SSS | P1 Fig. 3(c) | -3.7 | 44.4 |
| SSS | P2 Fig. 3(c) | -1.3 | 14.8 |
| SPR | $\lambda = 1.0, \alpha = 0.1$ | -2.3 | 45.9 |
| SPR | $\lambda = 1.3, \alpha = 0.3$ | -0.2 | 31.5 |
| MLA | ResNet-18 | -3.0 | 50.0 |
| MLA | ResNet-18 | -2.5 | 30.0 |
| SIM | Perspective | -1.0 | 48.1 |
| SIM | Perspective | -0.2 | 29.7 |

### 4.2 Training for robustness.

To test for robustness, we perform $\ell_\infty$ attacks using the fast gradient sign method (FGSM), presented by Goodfellow et al. (2015), on the FashionMNIST dataset (Xiao et al., 2017). We choose a 4-layer fully-connected network of size $784 \times 64 \times 32 \times 16 \times 10$, denoted as $\mathcal{N}_{fc}$, to construct the state matrices and outputs. A mini-batch of size 64 was used for training $\mathcal{N}_{fc}$, achieving an 80% test set accuracy[4].

FGSM generates adversarial examples, $\check{u}$, by taking a step of size $\epsilon$ in the direction of the sign of its gradient taken with respect to the input, $\check{u} = u + \epsilon \cdot \text{sgn}(\nabla \mathcal{N}_{fc}(u))$. In our experiments, we set $\epsilon = 0.004$ or $\epsilon = 0.008$. For each batch of the test set, we perturb 50% of pixels and leave 50% unperturbed. We evaluate model's robustness using the prediction accuracy on adversarial examples, $\mathbf{E}_{\check{u},y}(\mathbf{1}_{y=\text{sgn}(f(\check{u}))})$, which measures the ability of a model resisting them. Figure 4 illustrates the adversarial robustness with respect to different weight sparsity. In both cases, we observe that $\ell_1$-norm leads to a more robust model as compared to perspective relaxation, and continues to maintain robustness with approximately 45% fewer parameters. Moreover, the $\ell_1$-norm approach exhibits more robustness than the original

---

[4]The baseline model is trained on a single Nvidia Tesla K80 GPU

baseline network $\mathcal{N}_{fc}$ with higher model sparsity, until the sparsity reaches an over-sparsified threshold that leads to an inevitable capacity degradation.

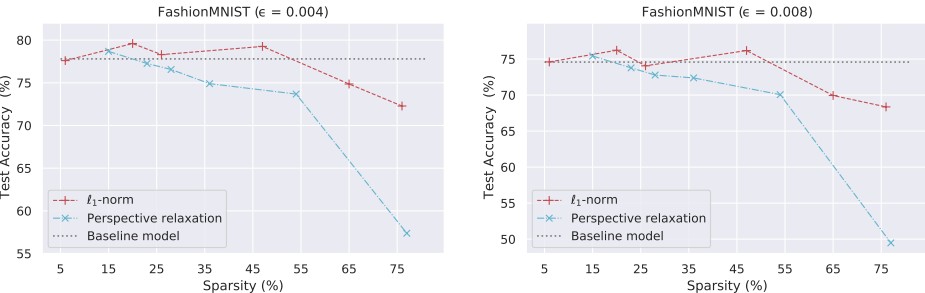

Figure 4: Trade-off curve for robustness with varying weight sparsity. The $\ell_1$-norm is more robust under adversarial attack than the perspective relaxation and the baseline model.

## 5 Related Work

Recent works (Bai et al., 2019; Chen et al., 2018; El Ghaoui et al., 2021; Winston & Kolter, 2020) have proposed an emerging "implicitly-defined" structure in deep learning, where the intermediate hidden states are defined via a "equilibrium" (fixed-point) equation, and the outputs are determined only implicitly by the equilibrium solution of such underlying equilibrium equation. Researchers have developed different classes of implicit models and demonstrated their potential in graph neural networks (Gu et al., 2020), differential equation models (Chen et al., 2018), physical control (de Avila Belbute-Peres et al., 2018), and many others (Amos et al., 2018). Recent work has shown that these implicitly defined models can be successfully sparsified, reducing their training and inference complexity (Ta et al., 2022). This leads to a more efficient model that operates in the same high-dimensional feature space but with a reduced representational complexity. One of the popular approaches in parameter reduction is to remove parameters with the smallest magnitude, a technique called *magnitude pruning* (Han et al., 2015; Zhu & Gupta, 2018; Molchanov et al., 2017). Magnitude pruning eliminates weights based on a learned magnitude or criterion of parameters with an *a priori* threshold (Yeom et al., 2019), which requires trial-and-error or heuristics. Others have considered a more principled way of determining the importance of parameters, including *structured pruning* (Sui et al., 2021; Chen et al., 2021) and *directional pruning* (Chao et al., 2020). Recent works have also view the problem from the perspective of optimization, such as *convex pruning* (Aghasi et al., 2020) or the *perspective reformulation technique* (Frangioni & Gentile, 2006; Atamtürk & Gómez, 2020). In addition, starting with Szegedy et al. (2014), a large number of works have shown that deep neural networks (DNNs) are vulnerable to adversarial samples (Goodfellow et al., 2015; Kurakin et al., 2017; Papernot et al., 2016a). The vulnerability of DNNs has motivated the study of building models that are robust to such perturbations (Madry et al., 2018; Papernot et al., 2016b; Raghunathan et al., 2018; Gowal et al., 2018). Defense strategies against adversarial examples have primarily focused on training with adversarial examples (Tramèr et al., 2017; Madry et al., 2018) or with a carefully designed penalty loss (Qin et al., 2019; Dhillon et al., 2018).

## 6 Conclusion

In this work, we present state-driven implicit modeling, a flexible convex optimization scheme for training an implicit model without expensive implicit differentiation, based on fixing the internal state and outputs from a given baseline model. We describe the convex training problem and parallel algorithms for training. By introducing an appropriate objective and setup, we demonstrate how state-driven implicit modeling can be applied to train sparse models that are consistently more robust under adversarial attacks. Our results validate the effectiveness of our approach and highlight promising directions for research that bring convex optimization, sparsity, and robustness-inducing techniques into implicit models.

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

## A  PROOFS

**Proof of Theorem 2.3**  We first prove the existence of a solution $x \in \mathbb{R}^n$ to the equation $x = \phi(Ax + b)$ if $\lambda_{\mathrm{pf}} < 1$. Since $\phi$ satisfies Assumption (2.1), we have that for $t \geq 1$, the picard iteration

$$x_{t+1} = \phi(Ax_t + b), \; x_0 = 0, \; t = 0, 1, \cdots$$

satisfies

$$
\begin{aligned}
|x_{t+1} - x_t| &= |\phi(Ax_t + b) - \phi(Ax_{t-1} + b)| \\
&\leq |A(x_t - x_{t-1})| \leq |A||x_t - x_{t-1}| \\
&\leq |A|^t |x_1 - x_0|.
\end{aligned}
$$

Hence, for every $t, \tau \geq 1$, we have

$$
|x_{t+\tau} - x_t| = \left| \sum_{i=t+1}^{t+\tau} (x_i - x_{i-1}) \right| \leq \sum_{k=t}^{t+\tau} |A|^k |x_1 - x_0|
$$

$$
\leq |A|^t \sum_{k=0}^{\tau} |A|^k |x_1 - x_0| \leq |A|^t \sum_{k=0}^{\infty} |A|^k |x_1 - x_0|
$$

$$
= |A|^t (I - |A|)^{-1} |x_1 - x_0|.
$$

The inverse of $I - |A|$ exists as $\lambda_{\mathrm{pf}}(|A|) < 1$. Since $\lim_{t \to \infty} |A|^t = 0$, we have

$$
0 \leq \lim_{t \to \infty} |x_{t+\tau} - x_t| \leq \lim_{t \to \infty} |A|^t (I - |A|)^{-1} |x_1 - x_0| = 0.
$$

We obtain that $x_t$ is a Cauchy sequence, and thus the sequence converges to some limit point, $x_\infty$, which by continuity of $\phi$ can be obtained by $x_\infty = \phi(Ax_\infty + b)$, thus establishes the existence of a solution to $x = \phi(Ax + b)$.

For uniqueness, consider two solutions $x_a, x_b \in \mathbb{R}^n_+$ to the equation, the following inequality holds,

$$
0 \leq |x_a - x_b| \leq |A||x_a - x_b| \leq |A|^k |x_a - x_b|.
$$

As $k \to \infty$, we have that $|A|^k \to 0$, and it follows that $x_a = x_b$, which establishes the unicity of the solution.

**Proof of Theorem 2.4**  Consider a neural network $\mathcal{N}$ in its equivalent implicit form $(A_{\mathcal{N}}, B_{\mathcal{N}}, C_{\mathcal{N}}, D_{\mathcal{N}}, \phi)$, since the matrix $|A_{\mathcal{N}}|$ is strictly upper triangular, all of its eigenvalues are zeros, automatically satisfying the PF sufficient condition for well-posedness. From the Collatz-Wielandt formula (Meyer, 2000), the PF eigenvalue of a well-posed implicit model can be represented as

$$
\lambda_{\mathrm{pf}}(|A|) = \inf_{s > 0} \left\| \mathbf{diag}(s)|A|\mathbf{diag}(s)^{-1} \right\|_\infty.
$$

The scaling factor $s$ such that $\left\| \mathbf{diag}(s)|A|\mathbf{diag}(s)^{-1} \right\|_\infty < 1$ can be obtained by solving

$$
s_i = 1 + \sum_{j=i+1}^{n} |A_{i,j}| s_j, \; i \in [n],
$$

which can then be solved by backward substitution.  The new model matrices $(A', B', C', D', \phi)$, are obtained by

$$
\left( \begin{array}{c|c} A' & B' \\ \hline C' & D' \end{array} \right) = \left( \begin{array}{c|c} SAS^{-1} & SB \\ \hline CS^{-1} & D \end{array} \right)
$$

where $S = \mathbf{diag}(s)$, with $s > 0$ a PF eigenvalue of $|A|$. More generally, provided that $\lambda_{\mathrm{pf}}(|A|) < 1$, we simply set $s = (I - |A|)^{-1}\mathbf{1}$, which can be obtained as the limit point of fixed-point iterations.

# B    MORE ON PARALLEL TRAINING

**Data structure.**    Fitting all the weight matrices into memory requires a substantial amount of storage space. However, we can leverage the high-sparsity property of the problem to reduce the memory consumption when storing the weight matrices. In the high-sparsity regime, schemes known from high-performance computing such as compressed sparse row (CSR) and compressed sparse column (CSC) can store indices of matrices, respectively. Since in this problem, we operate in a row-wise fashion, we choose to store the weight matrices in CSR format. CSR represents the indices in an $n = n_r \times n_c$ matrix using row and column index arrays. The row array is of length $n_r$ and store the offsets of each row in the value array in $\lceil \log_2 m \rceil$ bits, where $m$ is the number of non-zero elements. The column array is of length $m$ and stores the column indices of each value in $\lceil \log_2 n_c \rceil$ bits. The total storage space required is therefore $n_r \times \lceil \log_2 m \rceil + m \times \lceil \log_2 n_c \rceil$.

**Multiprocessing.**    Given state matrices from a neural network, the basis pursuit problem of (8) and (9) can be paralleled, each involving a single or a block of rows. Each block is trained independently by a child processor with an auxiliary objective, and returns the solutions back to the main processor. We implement our parallel training algorithm with the MULTIPROCESSING package using Python. The MULTIPROCESSING package[5] supports spawning processes and offers both local and remote concurrency. In Python, its Global Interpreter Lock (GIL) only allows one thread to be run at a time under the interpreter, which means we are unable to leverage the benefit of multi-threading. However, with multiprocessing, each process has its own interpreter and the instructions are executed by its own interpreter, which allows multiple processes to be run in parallel, side-stepping the GIL by using sub-processes instead of threads. In MULTIPROCESSING, a process is a program loaded into memory to run and does not share its memory with other processes. The decomposability of the training problem can be viewed as *data parallelism* where the execution of a function, i.e. solving the convex optimization problem, is parallelized, and the input values are distributed across processes. We use the `Pool` object to offer a means of defining a function in a module so that child processes can each import the module and execute it independently.

**Memory sharing.**    In MULTIPROCESSING, data in the arguments are pickled and passed to the child processors by default. In the basis pursuit problem, the state matrix $X$ and the input data matrix $U$ remain unchanged during task execution across all the processors, and thus only need read-only access to $X$ and $U$. Passing $X$ and $U$ to each processor whenever a new task is scheduled consumes a significant amount of memory space and increases the communication time. As a result, instead of treating them as data input to the function, we put $X$ and $U$ into a shared memory, providing direct access of the shared resources across processes.

**Ray.**    We also implement our parallel algorithm using RAY[6], an open-source and general-purpose distributed compute framework for machine learning and deep learning applications. By transforming the execution of the convex training problems into ray `actors`, we are able to distribute the input values to multiple ray actors to run on multiple ray nodes. Similar to the memory sharing in the multiprocessing approach, we use `ray.put()` to save objects into the ray object store, saving memory bandwidth by only passing the object ids around. We run our experiments on the Cori clusters[7] hosted by National Energy Research Scientific Computing (NERSC) Center and use the SLURM-RAY-CLUSTER scripts[8] for running multi-nodes.

**Performance benchmark.**    Figure 5 show the run-time for our serial and parallel implementation using both MULTIPROCESSING and RAY. We observe that MULTIPROCESSING

---

[5] https://docs.python.org/3/library/multiprocessing.html
[6] https://www.ray.io/
[7] https://docs.nersc.gov/systems/cori/
[8] https://github.com/NERSC/slurm-ray-cluster

provides the best speedup as compared to RAY. We hypothesize that since RAY is a general-purpose distributed compute framework, it contains more overhead than solving the training problem directly using MULTIPROCESSING.

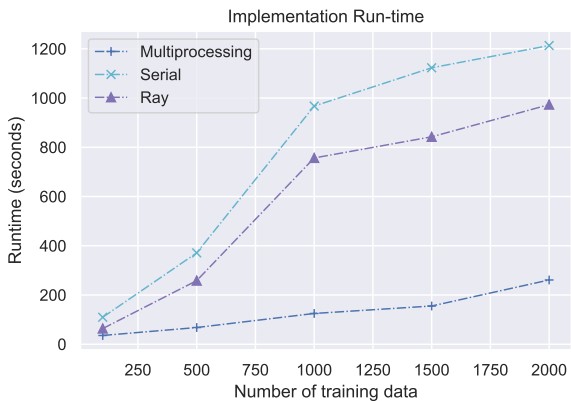

Figure 5: Performance benchmark for serial, multiprocessing (parallel), and Ray (parallel) implementation on FashionMNIST dataset using 8 processors.

## C   MORE ON NUMERICAL EXPERIMENTS

Table 2 and Table 3 shows the number of training samples, hyper-parameters, and adversarial test accuracy for perspective relaxation and $\ell_1$-norm objective functions with state and outputs matching penalties as in problem (7). For perceptive relaxation, we solve the following problem:

$$\min_{M,t,s} \; \alpha \sum_{ij} s_{ij} + \lambda_1 \left\| Z - (AX + BU) \right\|_F^2 \tag{10a}$$

$$+ \lambda_2 \left\| \hat{Y} - (CX + DU) \right\|_F^2 \tag{10b}$$

$$\text{s.t.} \quad (2d), \; t_{ij} \in [0,1], M_{ij}^2 \leq s_{ij} \cdot t_{ij}, \; s_{ij} \geq 0. \tag{10c}$$

For the $\ell_1$-norm problem, we solve the following problem:

$$\min_{M} \; \beta \sum_{ij} |M_{ij}| + \lambda_1 \left\| Z - (AX + BU) \right\|_F^2 \tag{11a}$$

$$+ \lambda_2 \left\| \hat{Y} - (CX + DU) \right\|_F^2 \; : \; (2d), \tag{11b}$$

where $\beta$ controls the degree of regularizing for robustness.

Table 2:  Experimental settings for perspective relaxation on Fashion-MNIST.

| | | | | | Test Acc. (%) | |
|---|---|---|---|---|---|---|
| # Train Samples | Sparsity (%) | $\lambda_1$ | $\lambda_2$ | $\alpha$ | $\epsilon = 0.004$ | $\epsilon = 0.008$ |
| 700 | 15 | 0.1 | 0.1 | 0.01 | 78.7 | 75.4 |
| 500 | 23 | 0.1 | 0.1 | 0.01 | 77.3 | 73.8 |
| 400 | 28 | 0.1 | 0.1 | 0.01 | 76.6 | 72.8 |
| 300 | 36 | 0.1 | 0.1 | 0.01 | 74.9 | 72.4 |
| 200 | 54 | 0.1 | 0.1 | 0.01 | 73.7 | 70.1 |
| 100 | 77 | 0.1 | 0.1 | 0.01 | 57.2 | 49.5 |

Table 3: Experimental settings for $\ell_1$-norm on Fashion-MNIST.

| | | | | | Test Acc. (%) | |
|---|---|---|---|---|---|---|
| # Train Samples | Sparsity (%) | $\lambda_1$ | $\lambda_2$ | $\beta$ | $\epsilon = 0.004$ | $\epsilon = 0.008$ |
| 600 | 20 | 0.1 | 0.1 | 0.001 | 79.6 | 76.2 |
| 1000 | 47 | 0.01 | 0.01 | 0.001 | 79.3 | 76.2 |
| 500 | 26 | 0.01 | 0.01 | 0.01 | 78.3 | 75.0 |
| 2000 | 6 | 0.01 | 0.01 | 0.01 | 77.6 | 74.6 |
| 900 | 65 | 0.01 | 0.01 | 0.001 | 74.9 | 69.9 |
| 400 | 76 | 0.01 | 0.01 | 0.001 | 72.3 | 68.4 |

