# OpenReview forum: "State-drive Implicit Modeling"
_ICLR.cc/2024/Conference — Submitted to ICLR 2024_

### Official Review · Reviewer_xzz3 · 2023-10-30

**Soundness:** 2 fair
**Presentation:** 3 good
**Contribution:** 1 poor
**Rating:** 3
**Confidence:** 3

**Summary:**

Derived from implicit models, the authors present State-driven Implicit Modeling (SIM), which is a method of training a regularized model with fewer parameters. Here, using the pre-trained or given model, SIM is formulated as a parallelizable convex optimization problem. The authors then show numerical experiments about how sparsity imposed in SIM affects the test accuracy and robustness.

**Strengths:**

The training process presented is straight-forward with the convex optimization problem. The results will be well reproducible.

**Weaknesses:**

The training method proposed is not a essentially learning an implicit model. As the training for layered neural network explicitly uses intermediate values, it is essentially estimating each layer of the NN with a regularized NN layer. To check the validity of the training process, its validity on implicit models should be presented, or at least on NN-approximated implicit models without using intermediate values.

Also, there is concern for the scalability of the training process. In the NN setting, the modeling $Z = AX + BU$ can be viewed as a skipping connection from input data to every intermediate layers in NN. If the dimension of the input $U$ becomes bigger and NN becomes deep, the size of the model architecture becomes too massive.

**Questions:**

1. Can SIM train other models with different task such as segmentation or generation?

2. Can you show the result on implicit model, which $X$ and $Z$ are not the intermediate values of NN networks?

---

### Official Review · Reviewer_JW8T · 2023-10-31

**Soundness:** 2 fair
**Presentation:** 2 fair
**Contribution:** 2 fair
**Rating:** 3
**Confidence:** 3

**Summary:**

This paper aims to mitigate the backward calculation in the training of implicit models. To achieve this goal, the authors first use a matrix to parameterize the update rule for the states in the implicit model. Then, they train the matrix to match the input and output for each layer in the original explicit/implicit model. As the convex property of the loss function, the optimization process is efficient. The authors claim that the trained model can be further applied to parameter reduction and robust training.

**Strengths:**

Using the state update rule to describe the implicit model is interesting. It is possible to use this framework to design new kinds of implicit models and may enlarge the application of implicit models.

**Weaknesses:**

1. The methodology in this paper mismatches the motivation of this research. In the abstract and introduction, the authors claim that the motivation of this work is to avoid the expensive differentiation for the backward propagation in the implicit model. Such a process usually happens during the training of the model. However, the method adopted in this paper requires a well-trained implicit model, which means that expensive differentiation has been performed. Thus, it is hard to say that this method can avoid the expensive backward differentiation in the implicit model.

2. As for the experiments, the reviewer believes some experimental settings are not common choices and hopes the authors can explain their reasons:

    (a)  In the sparsity experiment, the definition of sparsity is usually used in the unstructured pruning algorithm, which counts the number of non-zero parameters. However, all the baseline methods are structured pruning algorithms. In those algorithms, we care about the number of non-zero rows, rather than the non-zero elements. This difference in the definition makes the comparison unfair. Therefore, the reviewer suggests the author compare their method with the unstructured pruning algorithms.

   (b)  In the robustness experiment, the choice of $\epsilon$ is so small. On the Fashion MNIST datasets, the common choices of $\epsilon$ are 0.1 or 0.2. Also, the authors do not compare the proposed algorithm with any existing algorithm in the robustness experiment.

Due to the above problems, the reviewer suggests the author rethink the application of their framework and method. Although the reviewer believes the idea is interesting, the experimental results and the motivation in this paper are not strong enough.

**Questions:**

Listed in the weakness

---

### Official Review · Reviewer_jTGa · 2023-10-31

**Soundness:** 3 good
**Presentation:** 3 good
**Contribution:** 3 good
**Rating:** 6
**Confidence:** 2

**Summary:**

Training previous implicit models usually relies on expensive implicit differentiation for backward propagation. This work circumvent the costly backward computations by constraining the internal states and outputs to match that of a baseline model, which they name State-driven Implicit Modeling (SIM). Another advantage is that SIM can be implemented in a parallel manner. The author also demonstrates that SIM can be extended to parameter reduction and robust training.

**Strengths:**

1. The paper writing is crystal clear for people who doesn’t know implicit model a lot. It explains the concept clearly at the beginning and is consistent within the entire paper.
2. The idea of bypassing the expensive backward computation by matching the internal states and output is novel, as far as I know.
3. The experiments demonstrate that by using SIM, the model gains better performance in parameter reduction and under the adversarial attack.

**Weaknesses:**

1. Clarification on the experiments setting. For example, for a ResNet-20, is the X and Z constructed for every convolution layer as well as the final linear layer?
2. Missing comparison to other implicit modeling methods in the experiment of robust training.
3. The author states that, the training problem to find another well-posed model, with a desired task in mind, with the matching condition Y = MU, is “convex”, which is not intuitive during reading the main paper and also I don’t find anything in the supplementary about this as well.
4. “The corresponding state matrices X, Z will then be appropriately rescaled after running a single forward pass. ” here it seems like gamma is a hyper-parameter in the SIM scheme, but no ablation on this. Correct me if I am wrong.
5. Typo: figure 2 & 3 right side, the sparsity (%) is somehow occluded.

**Questions:**

1. What is the motivation of constraining the states and outputs of the implicit model to match those baseline states? Is this idea inspired by any previous work? Has any previous work that tries to use constraining internal states to represent a baseline model?

2. In Figure 4, why the sparsity increase and the L1-norm model gains better prediction accuracy under the attack? It’s between sparsity 15 to 25, and and 25 to 55.

---

### Official Review · Reviewer_437Q · 2023-10-31

**Soundness:** 3 good
**Presentation:** 2 fair
**Contribution:** 3 good
**Rating:** 5
**Confidence:** 3

**Summary:**

The authors propose a training scheme State-driven Implicit Modeling (SIM) that learns an implicit model by matching the dynamics from a pretrained implicit models. The authors use a convex training algorithm to overcome the costs associated with implicit differentiation. The approach is tested on parameter reduction and robustness experiments.

**Strengths:**

- The preliminaries and introduction are well written giving a good background on the motivation of the paper.
- The experiments demonstrates that SIM indeed provides improvements on the tasks tackled in the paper.

**Weaknesses:**

1. Claim that implicit models (Equilibrium Models in the context of this paper) rely on expensive implicit differentiation is incorrect. One of the main reasons they are used is the cheap Backward Pass -- Any Krylov Method can easily solve the problem, especially if used with [1] which reduces the condition number of the jacobian, the convergence is even faster
    * There are quite a few preprints on how to accelerate the Backward Pass but most of them are built on the incorrect premise that the backward pass is slow -- mostly because they use a Nonlinear Solver to solve a linear problem.
2. Experiments are on a small scale. Implicit models are extremely notorious for false positives on small scale -- see ablation studies in [2] and [3].
3. Robustness experiments don't have comparison to other methods

[1] Bai et. al. https://proceedings.mlr.press/v139/bai21b/bai21b.pdf (ICML 2021)

[2] Pal et. al. https://arxiv.org/pdf/2201.12240.pdf (IEEE HPEC 2023)

[3] Bai et. al. https://openreview.net/forum?id=B0oHOwT5ENL (ICLR 2022)

**Questions:**

1. If we compared SIM against Implicit Differentiation (with GMRES or other Krylov methods not a nonlinear solver like Broyden) how would the wall clock times compare?
2. Do we see the trends hold if we use larger datasets -- ImageNet for example?

---

### Meta-Review · Area_Chair_J9pC · 2023-12-06

**Metareview:**

There is no author rebuttal. All reviewers agree to reject the work.

**Justification For Why Not Higher Score:**

N/A

**Justification For Why Not Lower Score:**

N/A

---

### Decision · Program_Chairs · 2024-01-16

Reject